# Highly Birefringent and Low-Loss Hollow-Core Anti-Resonant Fiber Based on a Hybrid Guidance Mechanism

Xu'an Liu [1], Weixuan Luo [2,3], Xiaogang Jiang [4,5] and Bin Zhang [2,3,*]

1 School of Information Engineering, Huangshan University, Huangshan 245041, China; liuxan@hsu.edu.cn
2 Hangzhou Institute of Advanced Studies, Zhejiang Normal University, Hangzhou 311231, China
3 Key Laboratory of Optical Information Detection and Display Technology of Zhejiang, Zhejiang Normal University, Jinhua 321004, China
4 College of Teacher Education, Quzhou University, Quzhou 324000, China
5 Ningbo Innovation Center, Zhejiang University, Ningbo 315100, China
* Correspondence: binzhang@zjnu.edu.cn

**Abstract:** A highly birefringent and low-loss hollow-core anti-resonant fiber (HC-ARF) based on a hybrid guidance mechanism is proposed and investigated by using a finite element method. The hybrid guidance mechanism is caused by the anti-resonance effect and the total internal reflection effect. The proposed HC-ARF is obtained by employing twin symmetrical and mutually tangential elliptical arc anti-resonance layers (EA-ARLs) in a conventional 8-tube HC-ARF. Because of the anti-resonance mechanism and the total internal reflection mechanism in the EA-ARL, mode coupling appears between the core mode and the cladding mode. Simulation results indicate that the proposed HC-ARF can achieve birefringence as high as $10^{-2}$ in a near-infrared range of 1400 nm to 1600 nm and a low confinement loss (CL) of $7.74 \times 10^{-4}$ dB/m ($9.26 \times 10^{-4}$ dB/m) for x- and y-polarization components of the fundamental mode (FM) at 1550 nm. In addition, the existence of the 8-tube anti-resonance structure in the cladding significantly suppresses the CL of the x-polarization component of the FM significantly, but the impact on the CL of the y-polarization FM can be ignored, which is determined mainly by the twin EA-ARLs. Furthermore, the performance of the birefringence and CL are also investigated by changing the values of other fiber structure parameters. Our proposed structure successfully shows the ability of the hybrid guidance mechanism in the application of CL manipulation of orthogonal polarization components.

**Keywords:** optical fibers; anti-resonant layers; high birefringence; low confinement loss

## 1. Introduction

Highly birefringent fibers are widely used in optical communication systems, optical sensors, optical filters, and interference devices [1]. In the past decades, researchers have developed photonic crystal fibers (PCFs) with ultra-high birefringence based on total internal reflection light-guiding mechanisms. For example, ultra-high birefringence up to $10^{-2}$ at a wavelength of 1550 nm was obtained by constructing an asymmetric fiber core in a tiny region in the center of the PCF [2–7]. However, these solid-core highly birefringent PCFs have a some limitations, such as low damage threshold and significant absorption [8,9].

On the other hand, hollow-core fibers can confine optical light in its central air region in which the absorption losses are much lower than those in the dielectric media. The hollow-core fibers can also minimize the effect of the matrix material on light-guiding properties and significantly reduce fiber dispersion and optical nonlinearity, Therefore, the hollow-core fibers are considered as a promising alternative to PCFs [10,11].

The hollow-core fiber can be classified into two categories according to their light-guiding mechanism: the hollow-core photonic bandgap fiber (HC-PBGF) and the hollow-core anti-resonant fiber (HC-ARF). Photonic band gaps can be generated by using periodic

air hole arrays in the cladding, and HC-PBGF is based on photonic bandgap effect for confining light in the fiber core [12,13] in which the birefringence can be caused by using asymmetric cores [14–17]. Although HC-PBGF can achieve high birefringence and low loss, it still has the disadvantages of limited transmission bandwidth, low damage threshold, severe group velocity dispersion, and poor spatial mode purity [18]. The best-reported results, which show a phase birefringence of $2.5 \times 10^{-4}$ and a minimum loss of 4.9 dB/km, were achieved with a 19-cell PBGF designed by Fin et al. in 2014. However, its bandwidth was narrow, only 14 nm [17,19].

For HC-ARF, the light is guided by suppressing the coupling of core and cladding modes. Therefore, it has some optical characteristics, such as wide transmission bandwidth, extremely low loss, weaker group velocity dispersion, simple structure, and flexibility [20]. Furthermore, both light and matter are in the hollow core, resulting in a long interaction distance between light and matter, a compact structure, and a strong interaction between matter and fiber-guided light. Therefore, it can be used as a platform for lab-in-fiber and has important applications in mid-infrared laser [21] and mid-infrared gas detection [22]. In addition, it has potential application advantages in biomolecule detection based on the localized surface plasmon resonance (LSPR) effect. At present, fiber optic sensors based on LSPR usually use fiber splicing or form a tapered fiber structure to increase the interaction between the sensing biomolecules and the light wave to improve sensitivity when detecting some biological molecules, such as creatinine [23,24], cardiac troponin I [25,26], alanine aminotransferase [27], acetylcholine [28], aflatoxin B1 [29], and p-cresol [30,31]. By coating metal nanoparticles on the cladding tubes and using a syringe pump to fill the biomolecular analyte into the hollow core, the HC-ARF can be made into biosensors based on LSPR, which are expected to have potential advantages in biomolecule detection.

In recent years, researchers have made some progress in using anti-resonant fibers to design high-birefringence hollow-core fibers [32–41]. Currently, HC-ARFs are used mainly to achieve high birefringence by introducing asymmetry in its cladding tube, such as adjusting the thickness of the cladding tube in a certain polarization direction [1,9], using two nested tubes in a certain polarization direction [13], and nesting silicon layers in a certain polarization direction [32]. However, it is difficult for these fiber structures to obtain high birefringence, and the enhanced birefringence is still lower than $10^{-3}$. HC-ARF based on a hybrid guidance mechanism is considered as a solution to break through the birefringence limit. This type of fiber satisfies multiple light-guiding effects simultaneously, and we can design a highly birefringent fiber by using the advantage of different light-guiding mechanisms. Du et al. proposed a highly birefringent terahertz fiber with a birefringence higher than $10^{-2}$ and low loss of $1.69 \times 10^{-4}$ dB/cm ($9.14 \times 10^{-3}$ dB/cm) for the x(y)-polarization component of the fundamental mode (FM) at 1 THz, which is based on the hybrid guidance mechanism of the anti-resonant mechanism and the total internal reflection mechanism [42]. Guo et al. studied a highly birefringent HC-ARF based on a hybrid guidance mechanism of the photonic bandgap mechanism and the anti-resonant mechanism, and a birefringence of $3.91 \times 10^{-3}$ was achieved, while the loss was higher than $10^{-3}$ dB/km [10]. Research on hybrid waveguide-based hollow fiber is still relatively scarce, and only few superior properties have been found, which need to be further investigated.

In this study, a high birefringence and low-loss HC-ARF is proposed and investigated by using the finite-element method (FEM). The cladding of the HC-ARF is formed by a traditional pure 8-tube anti-resonance structure, and a pair of negative curvature and tangent elliptical arc anti-resonance layers (EA-ARLs) are employed in the hollow region. The twin EA-ARLs in our design introduce prominent structural asymmetry, which enhances the birefringence and decreases the confinement loss (CL) of the y-polarization component of the FM. Moreover, the 8-tube anti-resonant structure can significantly reduce the CL of the x-polarized FM, while the change of CL of the y-polarization component of the FM is negligible. Finally, the fiber properties are investigated by optimizing the structural parameters.

## 2. Fiber Structure and Performance Analysis

Figure 1 shows a cross-sectional schematic of the proposed HC-ARF which is composed of a pair of tangential negative curvature EA-ARLs in a typical pure 8-tube anti-resonant hollow structure. A waveguide structure is formed as a result of the interaction between the twin EA-ARLs. Thus, the proposed HC-ARF has a hybrid guidance mechanism that is combined with the anti-resonance effect and the total reflection effect, and it has features of high birefringence and low loss. As shown in Figure 1, $d$ is the diameter of the cladding tube, $t_1$ is its thickness, and $D$ is the diameter of the hollow circular. $t_2$ is the thickness of the EA-ARL, and the short and long semi-axes are denoted by $a$ and $b$, respectively. The ellipticity of the EA-ARL is denoted by $R$, and $R = b/a$. In order to analyze the birefringence and CL performance of the proposed HC-ARF, its mode and transmission constants are calculated by using the finite element analysis method. The HC-ARF structure parameters in Figure 1 are $D$ = 26 μm, $d$ = 12 μm, $t_1$ = 0.56 μm, $t_2$ = 0.85 μm, $b$ = 18 μm, and $R$ = 1.5. To save computational resources, an anisotropic perfectly matched layer with a thickness of 15 μm is placed on the outside of the cladding. The maximum element size of the glass region is $\lambda/10$, and that of the air region is $\lambda/8$. Thus, the total mesh elements of the model reach 446,840, and the rate of the effective refractive index change in more dense mesh set is negligible at 0%.

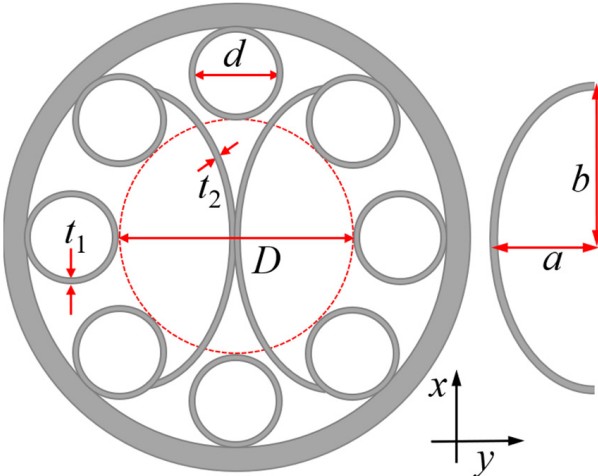

**Figure 1.** Cross section of the proposed HC-ARF.

The birefringence caused by the effective refractive index difference between the x- and y-polarization components of the FMs can be expressed as:

$$B = \left| n_y - n_x \right| \tag{1}$$

where $n_x$ and $n_y$ are the effective refractive indices of the $x$ and $y$ polarization components of FMs, respectively. The *CL* of the fiber transmission mode can be calculated by the following formula [43,44]:

$$CL = \frac{20}{\ln 10}\left(\frac{2\pi}{\lambda}\right)\mathrm{Im}(n_{eff}) \tag{2}$$

where $\mathrm{Im}(n_{eff})$ is the imaginary portion of the effective refractive index of the fiber polarization mode, and $\lambda$ is the operating wavelength.

The dependence of the birefringence and confinement loss on the tube thickness $t_1$ (0.2 μm to 1.8 μm) is analyzed. In our simulation, $D$ = 26 μm, $d$ = 12 μm, $t_2$ = 0.85 μm, $b$ = 18 μm, and $R$ = 1.5. Our operating wavelength is set at 1550 nm, unless otherwise stated. As shown in Figure 2, it can be seen that there are two low-loss windows according to the loss variation. This means the light is strictly confined to the hollow core in the anti-resonant region. On the contrary, the core mode has a good phase match with the cladding mode in the resonant region, which results in high CL value. In the first low-loss

window region, the lowest CL value ($7.74 \times 10^{-4}$ dB/m) was reached when the parameter $t_1$ is 0.56 μm. Therefore, the tube thickness $t_1$ is set as 0.56 μm for the following calculation.

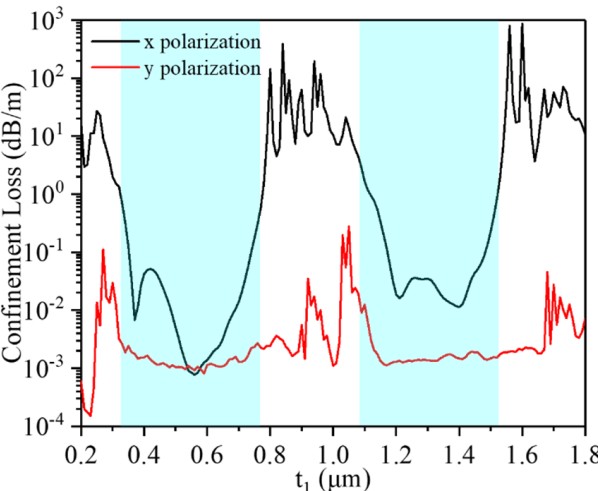

**Figure 2.** The dependence of CL of x− and y−polarized FMs on $t_1$.

To illustrate the function of the cladding 8-tube anti-resonant structure and the twin EA-ARLs, the birefringence characteristics between the proposed HC-ARF and the twin elliptical tube hollow-core anti-resonant fiber (ET-ARF) are analyzed. Furthermore, the differences of the CL characteristics among the twin ET-ARF, the pure 8-tube ARF, and the proposed HC-ARF are studied. Figure 3a shows two schematic diagrams of the cross-section of the twin ET-ARF and the pure 8-tube ARF. For comparison purposes, the dimensions and layer thicknesses of the elliptical ARL in both fibers are set as follows: $t_2 = 0.85$ μm, $b = 18$ μm, and $R = 1.5$. For the geometric parameters of the cladding 8-tube anti-resonant structure, $D = 26$ μm, $d = 12$ μm, and $t_1 = 0.56$ μm. Figure 3b shows the distribution of the transmitted electric field of the orthogonal polarization components of FMs of three types of fibers. In the fibers containing EA-ARL, the x- and y-polarization components of FMs are well-confined in the core, and most of them are concentrated in the center of the designed twin elliptical anti-resonant structure. Figure 3c shows the birefringence as a function of wavelength for the x- and y-polarization components of FMs. It can be seen that both types of fibers can obtain high birefringence up to $10^{-2}$ in the near-infrared band of 1400 nm to 1600 nm. In addition, one can see that the birefringence curve of the HC-ARF is almost identical to the birefringence curve of the ET-ARF, which means that the high birefringence is determined mainly by the twin tangent and symmetric EA-ARLs. Although the 8-tube anti-resonance structure in the cladding can limit the light to the hollow core and ensure low loss, it cannot provide high birefringence features. In contrast, the ET-ARF breaks the centrosymmetry and caused the refractive indices of the orthogonal polarization components of FMs to differ greatly from each. For example, the birefringence can reach 0.02314 for both types of fibers, but it is only $10^{-8}$ for the pure 8-tube anti-resonant fiber. As shown in Figure 3d, the CL of the y-polarization component of the FM of the ET-ARF is approximately $10^{-3}$ dB/m in the wavelength range of 1400 nm to 1600 nm, while CL of the x-polarization component is much higher. The CLs of the x- and y-polarization components of the FM for the ET-ARF are 13.29 dB/m and $7.38 \times 10^{-4}$ dB/m, respectively. They are $1.32 \times 10^{-3}$ dB/m and $1.04 \times 10^{-3}$ dB/m for the proposed HC-ARF and are both 0.46 dB/m for the pure 8-tube anti-resonant fiber. It is clear that the addition of the 8-tube anti-resonant structure significantly reduced the CL difference between the two polarization directions, and the x- polarization FM is reduced to the order of $10^{-3}$ dB/m, with a decrease of $10^5$ compared with the ET-ARF.

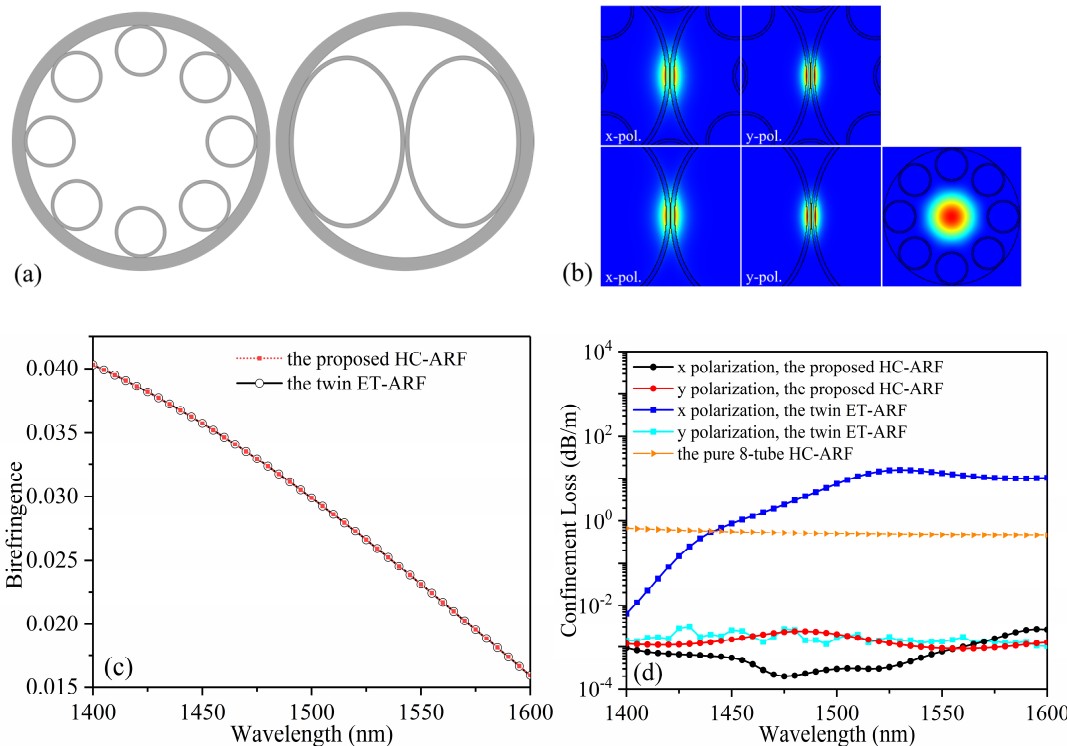

**Figure 3.** The comparison among the twin elliptical tube ARF, the pure 8−tube ARF, and the proposed HC−ARF. (**a**) Cross section of the pure 8−tube ARF and the twin elliptical tube ARF. (**b**) Comparison of electric field distribution. (**c**) Comparison of the birefringence in x− and y−polarization component. (**d**) Comparison of CL in x− and y−polarization component.

To explore the practical manufacturing redundancy of the proposed HC-ARF, the dependences of the birefringence and CL on parameters, $t_2$, $D$, and $d$, have been analyzed. As shown in Figure 4a, the birefringence increases by more than 180 times when $t_2$ varies from 0.75 μm to 1.0 μm. The insets in Figure 4a show the electric field distributions of the x- and y-polarization components of the FMs when $t_2 = 0.78$ μm. Compared with the results shown in Figure 3b, the size of the electric field is larger than the size of electric field in our proposed HC-ARF, which means that the performance of light confinement of the 8-tube anti-resonant structure and the twin EA-ARLs deteriorates. Figure 4(a) also shows that when $t_2$ is larger than 0.80 μm, the variations of the CL of both polarization components are approximately $10^{-3}$ dB/m, which means that the required hybrid waveguide mechanism is working. Birefringence increases with the increase of $t_2$. Both orthogonal polarization FMs have low CL regions, where the low CL region of the x-polarization FM is from 0.82 μm to 0.91 μm, and the low CL region of the y-polarization FM is from 0.80 μm to 0.85 μm. In the low-loss region, they all maintain a CL of $10^{-4}$ dB/m. In order to achieve high birefringence and low loss simultaneously, the optimal optimization parameter for $t_2$ is 0.85 μm.

Figure 4b,c show the birefringence and CL as a function of the diameter $d$ and $D$. The birefringence remains stable when the parameters d and $D$ are varied within our calculation range. For Figure 4c, the CL is approximately $10^{-3}$ dB/m for the x- and y-polarization components of the FM for $d$ in the range of 11 μm to 16 μm. The loss of two orthogonal polarization FMs is the lowest at $d = 12$ μm, and both remain at $10^{-4}$ dB/m. For Figure 4c, the CL fluctuates in the order of $10^{-3}$ dB/m as $D$ varies from 25 μm to 26.5 μm. When $D = 26$ μm, the loss of the two orthogonal polarization FMs is the lowest, both maintained at $10^{-4}$ dB/m. When $D$ is larger than 26.5 μm, the loss increases sharply. This means that the hollow core area is larger, which leads to a mismatch of the suppression effects on the CL between each waveguide mechanism.

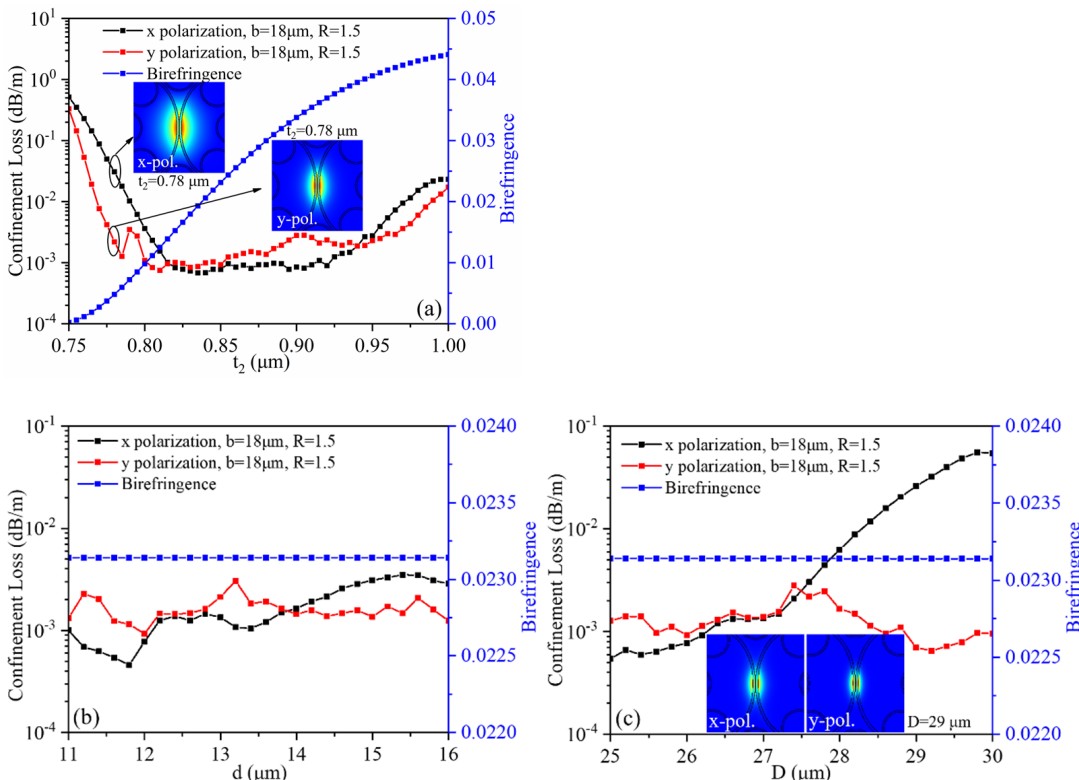

**Figure 4.** Birefringence and CL of the x− and y−polarized FMs as a function of the (**a**) thickness of EA−ARL $t_2$, (**b**) diameter of hollow circular region $D$, and (**c**) tube diameter $d$.

To further explore the performance of our proposed HC-ARF, the properties of the fiber were discussed by changing the separation distance of the twin EA-ARLs and the ellipticity of the EA-ARL. Figure 5 shows the birefringence and CL variation of the proposed HC-ARF, in which the ellipticity $R$ of the EA-ARL is varied in the range from 1.0 to 1.8. The birefringence increases with the increase of the ellipticity, and after the ellipticity $R > 1.2$, the increase of the birefringence became increasingly smaller. The decrease of the FM CL in the y-polarization component is less than one order of magnitude, while the decrease of the CL in the x-polarization component is larger than six orders of magnitude. When $R$ is 1.6, the CL value reaches its lowest of $1.85 \times 10^{-4}$ dB/m, but the CL in the y-polarization component increases to $10^{-3}$ dB/m. When $R$ is 1.5, both orthogonal polarization FM CLs remain at $10^{-4}$ dB/m. Based on the principle of maintaining high birefringence and low loss in both polarization fundamental modes of the proposed HC-ARF, the optimal parameter of $R$ is set to 1.5.

Figure 6a provides a schematic diagram of the structure of the HC-ARF. $H_1$ is the separation distance of the twin EA-ARLs. Figure 6b shows the birefringence and CL of the FMs as a function of $H_1$ based on the proposed HC-ARF. Figure 6c shows the electric field distribution of the x- and y-polarization components of the FMs when $H_1$ is taken as 1 μm, 2 μm, 3 μm, and 4 μm, respectively. We can see that the birefringence of the proposed HC-ARF decreases as $H_1$ increases. The CL of the x-polarized FM increases sharply due to the large amount of light energy being leaked to the cladding along the EA-ARL, indicating that the limiting effect of the 8-tube anti-resonant structure on the light energy of the x-polarized FM diminishes sharply when the pair of EA-ARLs are separated along the x-axis. On the other hand, the property of the CL of the y-polarization component of the FM is more complicated. With the increase of $H_1$, the CL of the y-polarized FM decreases. When $H_1$ is larger than 1 μm, more light energy is coupled into the cladding mode when the core mode and the cladding mode interact with each other, leading to an increase in the loss of the core mode.

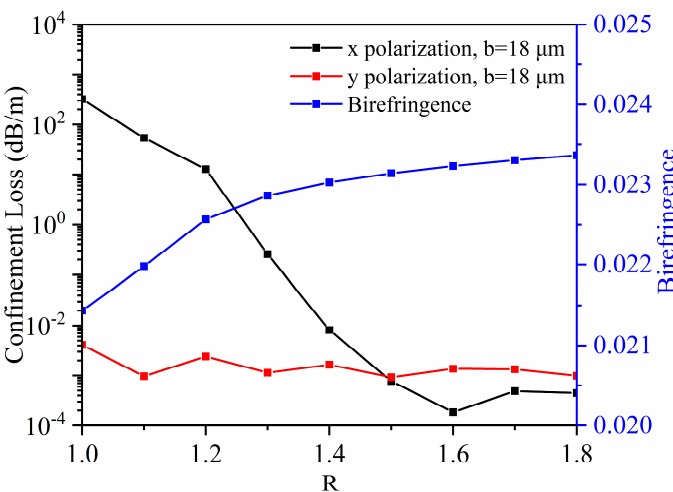

**Figure 5.** The CL and birefringence vs *R*.

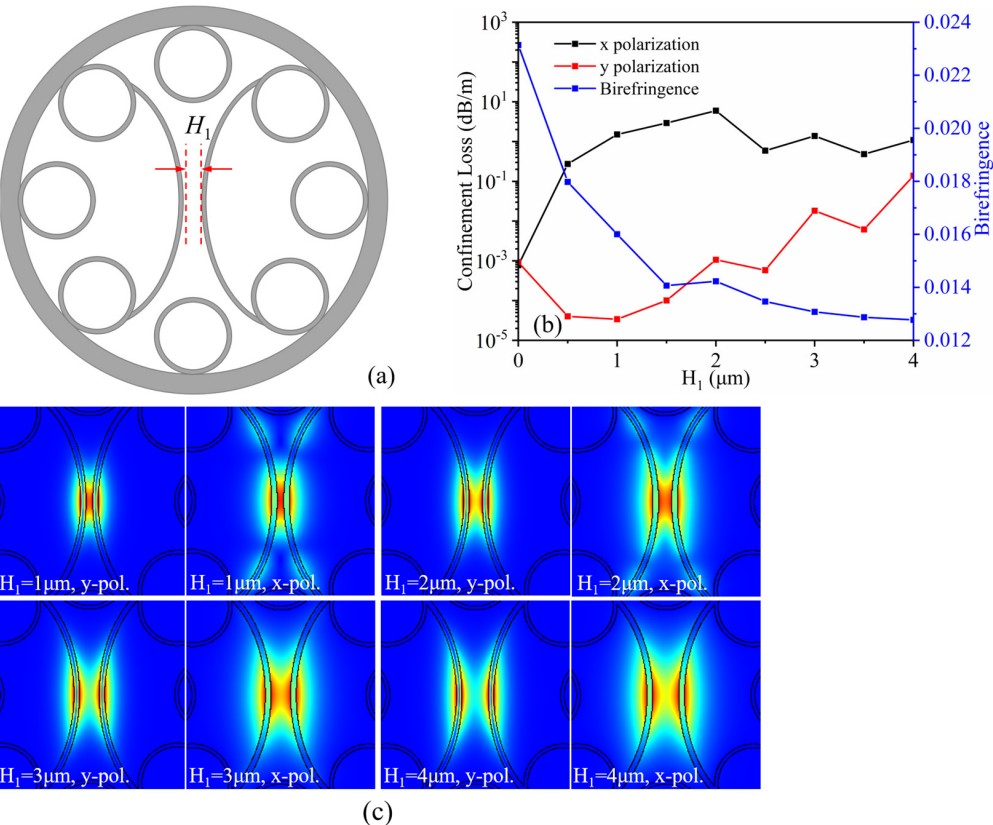

**Figure 6.** (**a**) Schematic cross section of proposed HC-ARF in which a pair of EA−ARLs are separate. (**b**) Birefringence and CL of the x− and y−polarization components as a function of distance $H_1$. (**c**) Electric field distributions of the x− and y−polarization components of the FMs at $H_1 = 1$ μm, $H_1 = 2$ μm, $H_1 = 3$ μm, and $H_1 = 4$ μm, respectively.

We also studied the birefringence and CL characteristics of the two orthogonal polarization FMs of the proposed HC-ARF when the twin EA-ARLs partially overlap. As shown in Figure 7a, we let the width of the overlap of two elliptical arc structures be $H_2$. We can see that when two elliptical arcs overlap, the structure of the fiber core undergoes a fundamental change, and the overlapping part forms a solid silica core, which affects the transmission performance of the fiber. Figure 7b shows the birefringence and CL of the FMs as a function of $H_2$ based on the proposed HC-ARF. When two elliptical arcs overlap, the

CL of the two orthogonal polarization FMs significantly increases. This is because silica is a high refractive index medium relative to air. Thus, the change of the size of the solid fiber core will seriously affect the anti-resonance period of the cladding anti-resonance tubes, resulting in a significant reduction of the CL suppression effect of the anti-resonance tubes on the two orthogonal FMs, and the two elliptical arc structures itself cannot effectively suppress the light field. As shown in Figure 7c, in the ET-ARF, when two elliptical arcs partially overlap, a large amount of energy leaks from the core along the elliptical tube. Therefore, in order to achieve high birefringence and low CL performance of the proposed HC-ARF, it is most appropriate to maintain tangency between the two EA-ARLs.

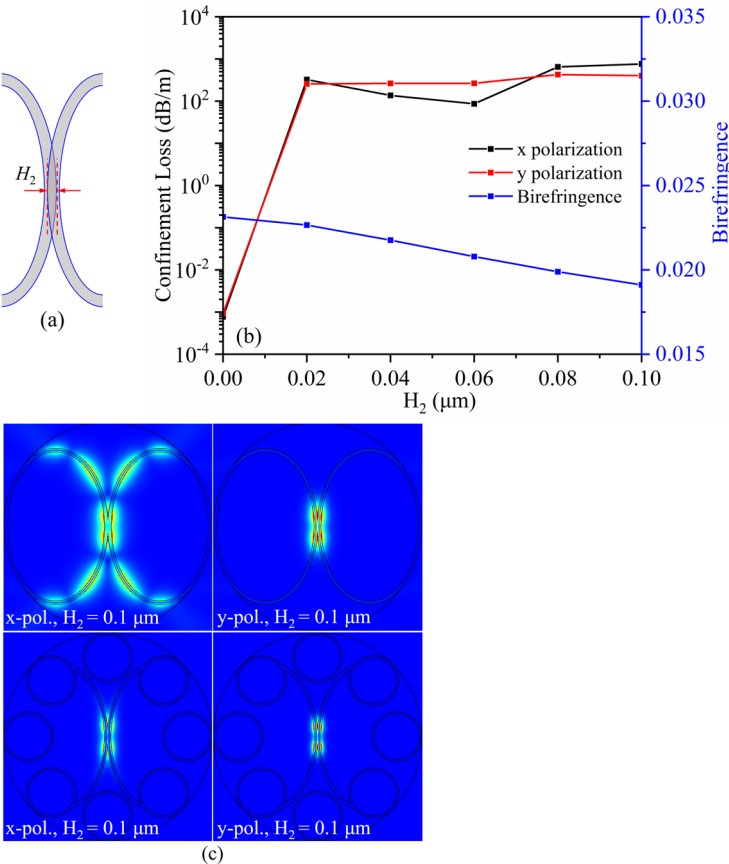

**Figure 7.** (**a**) Schematic cross section of proposed HC−ARF in which a pair of EA−ARLs are overlapping. (**b**) Birefringence and CL of the x− and y−polarization components as a function of distance $H_2$. (**c**) Electric field distributions of the two orthogonal polarization FMs for two types of fiber at $H_2 = 0.1$ μm.

Based on the above research, we can see that there is an optimal parameter value to simultaneously obtain high birefringence and minimum CL. The optimized structure parameters are $D = 26$ μm, $d = 12$ μm, $t_1 = 0.56$ μm, $t_2 = 0.85$ μm, $b = 18$ μm, and $R = 1.5$, while the two EA-ARLs remain tangent to each other. Under these conditions, the birefringence of the proposed HC-ARF can reach 0.231 with the x-polarization FM CL of $7.74 \times 10^{-4}$ dB/m and y-polarization FM CL of $9.26 \times 10^{-4}$ dB/m at 1550 nm. On the premise of not pursuing low loss and ensuring that the two polarization FM CLs are less than 0.01 dB/m, when the structural parameters of the fiber are $D = 26$ μm, $d = 12$ μm, $t_1 = 0.56$ μm, $t_2 = 0.97$ μm, $b = 18$ μm, and $R = 1.5$, the birefringence of proposed HC-ARF can reach 0.042 with the x-polarization FM CL of 0.0094 dB/m and y-polarization FM CL of 0.0036 dB/m at 1550 nm.

To facilitate a comparison, we summarize the performances of the reported high birefringent micro-structured fibers and make a comparison with our work, as shown

in Table 1. Up until now, the highest birefringence micro-structured fibers based on hollow core reported is up to $10^{-3}$ at 1550 nm in Ref. [10]. Our structure has higher order birefringence and has increased by one order of magnitude. Compared with the high birefringent micro-structured fibers based on solid core, our structure has the same order in birefringence.

**Table 1.** Comparison of the proposed HC-ARF with other published works.

| Reference | Light Guiding Mechanism | Type of Fiber Core | Wavelength | Birefringence | Minimum CL |
|---|---|---|---|---|---|
| [2] | | | 1.55 μm | $2.1 \times 10^{-2}$ | $9.10 \times 10^{-5}$ dB/m |
| [3] | | | 1.55 μm | $2.2 \times 10^{-2}$ | $<10^{-6}$ dB/m |
| [4] | Index-guiding mechanism | Solid core | 1.55 μm | $1.7 \times 10^{-2}$ | ~$1.0 \times 10^{-8}$ dB/m |
| [5] | | | 1.55 μm | $1.46 \times 10^{-2}$ | $6.10 \times 10^{-6}$ dB/m |
| [6] | | | 1.55 μm | $3.11 \times 10^{-2}$ | — |
| [7] | | | 1.55 μm | $2.02 \times 10^{-2}$ | $9.67 \times 10^{-5}$ dB/m |
| [14] | PBG-guiding mechanism | Hollow core | 1.55 μm | $<1.0 \times 10^{-3}$ | — |
| [17] | | | 1.533 μm | $2.5 \times 10^{-4}$ | <0.01 dB/m |
| [1] | | | 1.55 μm | $1.4 \times 10^{-4}$ | 0.075 dB/m |
| [9] | AR-guiding mechanism | Hollow core | 1.55 μm | ~$1.5 \times 10^{-4}$ | 0.04 dB/m |
| [13] | | | 1.55 μm | $1.3 \times 10^{-5}$ | 0.02 dB/m |
| [19] | | | 1.55 μm | $3.07 \times 10^{-4}$ | $8.90 \times 10^{-4}$ dB/m |
| [10] | Hybrid-guiding mechanism | Hollow core | 1.55 μm | $3.35 \times 10^{-3}$ | $6.20 \times 10^{-5}$ dB/m |
| This work | | | 1.55 μm | $2.31 \times 10^{-2}$ | $1.85 \times 10^{-4}$ dB/m |

## 3. Discussion and Conclusions

For optical fiber fabrication, because our proposed fiber structure is based mainly on the HC-ARF, the preform can be obtained by using the stack-and-draw technique [45]. The main fabrication challenge is the size and position control of the pair of EA-ARLs because they need to be strictly tangent to each other during the fabrication process. However, as shown in Figure 1, we have seen some research teams report on the manufacturing methods of elliptical arc structures [18,34,46–51], which may be improved to manufacture the hollow-core fibers designed in our study. The curvature of the EA-ARL can be obtained by precisely controlling the tension [52].

In summary, we proposed a novel HC-ARF based on a hybrid guidance mechanism, caused by both the anti-resonance effect and the total reflection effect, which achieves ultra-high high birefringence and low FM loss. By employing a pair of tangent EA-ARLs into the hollow core of a pure 8-tube HC-ARF, the reported design achieves high birefringence and low CL performances. Simulation results confirm that an ultra-high birefringence up to the order of $10^{-2}$ is realized in the wavelength ranging from 1400 nm to 1600 nm, while the CL difference between the x- and y-polarization components of the FM is very small. The high birefringence can reach $2.31 \times 10^{-2}$, and the confinement loss is only $7.74 \times 10^{-4}$ dB/m ($9.26 \times 10^{-4}$ dB/m) for the x-polarization (y-polarization) component of the FM. It is a significant enhancement in the birefringence compared with the previous reports. Furthermore, our proposed structure successfully shows the ability of the hybrid guidance mechanism in the application of CL manipulation of orthogonal polarization components. The proposed high birefringence HC-ARF can be used to construct interference spectra based on the Sagnac interference ring for sensing detection, such as refractive index-related biosensing, gas sensing, and sensing of fiber pressure, bending, temperature, and other parameters that may cause refractive index changes.

**Author Contributions:** Conceptualization, X.L., X.J. and B.Z.; methodology, X.L. and X.J.; investigation, X.L. and W.L.; resources, X.L., W.L., X.J. and B.Z.; data curation, X.J. and B.Z.; writing—original draft preparation, X.L.; writing—review and editing, X.L., X.J. and B.Z.; supervision, X.L. and B.Z.; project administration, X.L.; funding acquisition, X.L., X.J. and B.Z. All authors have read and agreed to the published version of the manuscript.

**Funding:** This research was funded by the Natural Science Research Project of Anhui Province Education Department (No. 2022AH051958), the Start-up Fund of Huangshan University (No. 2020xkjq015), "Pioneer" and "Leading Goose" R&D Program of Zhejiang (No. 2022C03084, No. 2022C03066), Natural Science Foundation of China (No. 62205296, No. 62205297), and NingboTech University (No. 20201203Z0189).

**Institutional Review Board Statement:** Not applicable.

**Informed Consent Statement:** Not applicable.

**Data Availability Statement:** All data are provided in full in the results section of this paper.

**Acknowledgments:** The authors would like to thank their supervisors and colleagues who have provided much assistance with the research.

**Conflicts of Interest:** The authors declare no conflict of interest.

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
