# Peer review of "Highly Birefringent and Low-Loss Hollow-Core Anti-Resonant Fiber Based on a Hybrid Guidance Mechanism"

_photonics, doi:10.3390/photonics10050525_

Round 1

Reviewer 1 Report

Dear Editor,

The authors propose a novel design of a birefringent and low loss hollow-core antiresonant fiber that can support a hybrid guidance mechanism of antiresonance and conventional total internal reflection. The authors numerically identified that the proposed design can provide a birefringent as high as 10-2 in the near-infrared spectral domain with 7-7x10-4 dB/m confinement loss.

In general, the manuscript is well written and the numerical results sound consistent. However, the current version of the manuscript needs further improvements before considered for publication in the Photonics journal. In particular:

In the introduction, the authors describe the difference between bandgap and anti-resonance guidance mechanism in hollow-core fibers. I suggest to refer also to the review article on the different guidance mechanisms in such fibers (C. Markos et al. Reviews of Modern Physics 89, 045003, 4 (2017)).

Furthermore, the authors briefly overview the important properties of hollow-core fibers such as bandwidth, extremely low loss, dispersion, etc.. However, they should also importantly refer to the ability of these fibers for applications in the mid-infrared spectral domain (Y. Wang, et al. Opt. Lett. 45, 1938-1941 (2020) and Sci. Rep 11, 3512 (2021)).

The authors briefly refer to the experimental feasibility of their design. However, it is not clear how in terms of fabrication the proposed fiber will maintain its semi-spherical design will be maintained during drawing given that tension can dramatically deform the inner structure.

Did the authors include the surface scattering and material loss in their calculations and if so, they should elaborate on what values they chose.

Overall, the current manuscript can be accepted for publication, if the authors address the aforementioned comments.   

Author Response

Dear Editors and Reviewers:

Thank you very much for your evaluation and comments on our paper. We have studied the comments carefully and made revisions one by one in the revised manuscript according to your suggestions. We have tried our best to revise our manuscript according to your kind advices. Attached please find the revised version and response letter, which we would like to submit for your kind consideration.

We hope that these revisions are satisfactory and that the revised version will be acceptable for publication in Photonics.

We thank the reviewers again for their careful reading and helpful comments, which have improved our manuscript apparently.

Sincerely,

Xu’an Liu

School of Information Engineering, Huangshan University, Huangshan 245041, China  (E-mail: [email protected])

Bin Zhang

Hangzhou Institute of Advanced Studies, Zhejiang Normal University, Hangzhou 311231, China(E-mail:[email protected])

Reviewer 2 Report

The authors proposed a novel high birefringence fiber by combining the 8-tube anti-resonance structure and two asymmetrical elliptical arc anti-resonance layers. Overall, the manuscript is well presented, and the simulation results support the conclusions drawn. The research is well-conducted, and the manuscript is well-organized. I would recommend publication after the following revision:

The structural asymmetry introduced by a pair of elliptical arcs is the most challenging part for fabrication. The authors mentioned some references to address the fabrication challenges. More discussion fabrication tolerance study should be added. For example, the dependency on the arcs’ separation H was discussed from 0 ~ 4um. How about when the two arcs partially overlap with each other (distance < 0)?

Minor issue: plot of the twin ET-ARF in fig 3(c) is missing.

Author Response

(The authors gave the same response as above.)

Reviewer 3 Report

The authors of this study propose an HC-ARF based on a hybrid guidance mechanism that combines the anti-resonance effect with the total reflection effect to achieve ultra-high birefringence and low FM loss. It should undergo revision before this work can be published in a scientific journal. 

1.     Which kind of FEM based FEM-based commercial software has been used for simulation purpose? Authors should mention the essential parameters. 

2.     Authors should discuss  the other optical fiber based sensing work that should be relevant to proposed work for the detection of several biomolecules, including creatinine, cardiac troponin I, alanine aminotransferase, acetylcholine, aflatoxin B1, aflatoxin B1, and p-cresol biosensors based on localized surface plasmon resonance principle. This will help to know the application of proposed study. 

3.     In section 2. Fiber structure and performance analysis, authors should add the fiber dimension optimization results. Also, need to mention the dimension parameters in Fig. 1. 

4.     In Fig. 2, dependence of CL of x- and y-polarized FMs on t1. What is the reason for multiple peaks?

5.     As per above comments, authors should show some sensing applications of proposed fiber. 

6.     Authors should also add one section of fabrication and feasibility analysis of proposed work. 

7.     Authors should compare the performance of proposed work with existing works in a tabular form and add before conclusion section. 

Author Response

(The authors gave the same response as above.)

Round 2

Reviewer 3 Report

Revision is not satisfactory. Authors should incorporate the reviewer's suggestions in the paper. Few response is not as per reviewer's comments. Also, response is not incorporated in to the revised manuscript. 

Here is the detailed response.
1. There is no response to comment 2 in the revised paper. In addition, there is no appropriate response.
2. According to comment 3, the authors have not supplied the results of fiber dimension optimization.
3. According to comment 5, the authors of this paper have not conducted any research or presented any sensing results. They have discussed previous works in sensing. Also, the response was not incorporated into the revised manuscript.
4. According to comment 7, table 1 is unsatisfactory. Additionally, authors should compare a few more parameters.

Author Response

Dear Editors and Reviewers:

Thank you very much for your evaluation and comments on our paper. We have studied your feedback carefully and made revisions one by one in the revised manuscript according to your suggestions. We have tried our best to revise our manuscript according to your kind advices. Attached please find the revised version and response letter, which we would like to submit for your kind consideration.

We hope that these revisions are satisfactory and that the revised version will be acceptable for publication in Photonics.

We thank the reviewers again for their careful reading and helpful comments, which have improved our manuscript apparently.

Sincerely,

Xu’an Liu

School of Information Engineering, Huangshan University, Huangshan 245041, China  (E-mail: [email protected])

Bin Zhang

Hangzhou Institute of Advanced Studies, Zhejiang Normal University, Hangzhou 311231, China(E-mail:[email protected])

Round 3

Reviewer 3 Report

Satisfactory revision.